# Adolescent Socioeconomic Status and Mental Health Inequalities in the Netherlands, 2001–2017

**DOI:** 10.3390/ijerph16193605

**Published:** 2019-09-26

**Authors:** Dominic Weinberg, Gonneke W. J. M. Stevens, Elisa L. Duinhof, Catrin Finkenauer

**Affiliations:** 1Department of Interdisciplinary Social Science, Faculty of Social and Behavioural Sciences, Utrecht University, Utrecht 3584 CH, The Netherlands; 2Department of Biological Psychology, Vrije Universiteit Amsterdam, Amsterdam 1081 BT, The Netherlands

**Keywords:** socioeconomic status (SES), adolescent mental health problems subjective SES, adolescent educational level, health trends, health inequalities

## Abstract

Even in wealthy countries there are substantial socioeconomic inequalities in adolescent mental health. Socioeconomic status (SES) indicators—parental SES, adolescent subjective SES and adolescent educational level—are negatively associated with adolescent mental health problems, but little is known about the interplay between these SES indicators and whether associations have changed over time. Using data from the Dutch Health Behaviour in School-Aged Children (HBSC) studies (*n* = 27,020) between 2001 and 2017, we examined associations between three SES indicators and six indicators of adolescent mental health problems. Linear regressions revealed that adolescent subjective SES and adolescent educational level were independently negatively associated with adolescent mental health problems and positively associated with adolescent life satisfaction, but parental SES had negligible independent associations with adolescent mental health problems and life satisfaction. However, when interactions between SES indicators were considered, high adolescent subjective SES was shown to buffer the negative association between parental SES and adolescent mental health problems and the positive association between parental SES and life satisfaction. Despite societal changes between 2001 and 2017, socioeconomic inequalities in adolescent mental health were stable during this period. Findings suggest that all three SES indicators—parental SES, adolescent subjective SES and adolescent educational level—are important for studying socioeconomic inequalities in adolescent mental health.

## 1. Introduction

Even in relatively wealthy and egalitarian countries, such as the Netherlands, there are socioeconomic inequalities in adolescent mental health: adolescents with a lower socioeconomic status (SES) have more mental health problems than their higher SES peers [1,2,3]. To date, most studies on socioeconomic inequalities in adolescent mental health have measured parental SES, for example, parental affluence, educational level or occupation [3,4,5]. However, this strong focus on parental SES might have led researchers to underestimate the importance of two other indicators of healthcare SES—adolescent subjective SES and adolescent educational level—which may have stronger associations with adolescent mental health [3,6,7,8], or may moderate the association between parental SES and adolescent mental health. Furthermore, societal changes during recent decades may have affected the size of SES inequalities in adolescent mental health [9,10,11], but studies have rarely investigated changes in these inequalities over time. Therefore, this cross-sectional study uses a time-sequential design to investigate the interplay between three SES indicators—parental SES, adolescent subjective SES and adolescent educational level—and adolescent mental health problems and explores changes in these associations during the last two decades.

### 1.1. Associations between SES Indicators and Adolescent Mental Health Problems

Researchers have consistently found that adolescents with lower parental SES [3,4,5], lower subjective SES [8] and a lower educational level [6,12,13] have more mental health problems compared to their peers with higher levels on these SES indicators. Different, though overlapping, mechanisms have been proposed for these three associations. Lower parental SES has been found to have associations with adolescent mental health problems through mechanisms such as material hardship, harsher parenting, greater parental stress and parental mental health problems [4,5,14]. Adolescent subjective SES taps adolescents’ perception of the relative position of their family in the socioeconomic hierarchy and is theorised to influence mental health through social comparison effects [15,16,17,18]. More specifically, lower subjective SES may be associated with adolescent mental health problems through mechanisms such as feelings of resentment and shame and lacking psychosocial resources (e.g., optimism, coping strategies and perceptions of personal control and social support) that could help alleviate stressors [19,20,21]. Adolescent educational level indicates adolescents’ increasing agency and growing independence from their parents, including the establishment of their own SES [12,22]. A lower adolescent educational level may be associated with more mental health problems through mechanisms such as experiencing a negative classroom climate, with unsupportive, misbehaving peers [23] or perceiving that one’s educational position indicates a lack of effort or intelligence, which may generate feelings of academic inadequacy and inferiority [13,24,25,26]. 

Research on adolescent mental health problems has found that associations with the three SES indicators are not independent. Negative associations between parental SES and adolescent mental health problems generally weaken substantially when including adolescent subjective SES [27,28,29], though some studies found that these associations do not fully disappear [7,8,30]. Similarly, negative associations between parental SES and adolescent mental health problems weaken substantially when including adolescent educational level [6,13,31]. To our knowledge, only one study, of Slovenian adolescents, has included all three SES indicators simultaneously; the study found that adolescent subjective SES and adolescent educational level, but not parental SES, were negatively associated with adolescent mental health problems [32]. Extending the existing research, we investigate the independence and relative strength of associations between the three SES indicators and adolescent mental health problems.

### 1.2. A Moderating Role for Adolescent Subjective SES and Adolescent Educational Level?

Existing research has not explored the interplay between the different SES indicators. Adolescent subjective SES may moderate the negative association between parental SES and adolescent mental health problems. In a wealthy country like the Netherlands, with a strong welfare state that redistributes income and provides extensive public services [33], psychosocial mechanisms may explain much of the negative association between parental SES and adolescent mental health problems [15,16]. Therefore, in the presence of low parental SES, higher adolescent subjective SES may act as a protective factor, weakening the negative association between parental SES and adolescent mental health problems [21]. 

Adolescent educational level may also moderate the association between parental SES and adolescent mental health problems. On the one hand, according to the resource substitution theory, a higher adolescent educational level may attenuate the negative association between parental SES and adolescent mental health problems [34]. Adolescents in higher educational levels gain cognitive and psychosocial skills, such as problem-solving skills, which can be used to protect themselves from the harmful effects of low parental SES on their mental health [34,35]. On the other hand, the negative association between parental SES and adolescent mental health problems may be stronger for adolescents with a higher educational level. Adolescents who come from a lower SES family may not feel at home in a higher educational levels [36,37]. Such experiences may lead to feelings of social isolation, which are associated with mental health problems [38]. To our knowledge, no research has yet examined whether the association between parental SES and adolescent mental health problems depends upon adolescent subjective SES or adolescent educational level. 

### 1.3. Changes in Socioeconomic Inequalities in Adolescent Mental Health 

Several societal changes, both worldwide and specific to the Netherlands, may have caused changes in socioeconomic inequalities in adolescent mental health during the last two decades [11,39]. Firstly, the Great Recession, which lasted from roughly 2008 to 2013, substantially altered the socioeconomic environment by increasing material hardship, unemployment and financial uncertainty [40]. These changes may have increased parents’ and adolescents’ stress about future financial circumstances, especially among adolescents with lower parental SES, which increases the risk of adolescent mental health problems [41]. Secondly, between 2003 and 2017, there was a substantial reduction in student numbers in the lower educational levels in the Netherlands (adolescents are tracked into one of four educational trajectories at age 11–12) [42]. Based on its presumed negative influence on future educational and career options, the reputation of the lower educational levels has declined [43,44]. This may have motivated parents and adolescents to avoid the lower educational levels. As a consequence, these lower educational levels may have increasingly included students with relatively lower cognitive and psychosocial skills and worse mental health [43]. Thirdly, the last two decades have seen extraordinary changes in access to, and use of, digital media [45,46]. There is evidence that adolescents with lower parental SES have been most susceptible to the negative aspects of digital media, such as negative online interactions, and these adolescents have had an increased risk of mental health problems [47]. As such, existing socioeconomic inequalities in adolescent mental health may have been amplified during the last two decades.

Only a few studies have looked at changes over time in these socioeconomic inequalities in the Netherlands. Researchers studying European adolescents between 2002 and 2010 found an increase in socioeconomic inequalities, indicated by parental SES in psychosomatic complaints and a decrease in socioeconomic inequalities in life satisfaction [1]. Other studies have found stable socioeconomic inequalities in adolescent mental health in the Netherlands when using measures of adolescent subjective SES (between 2002 and 2010) [48] and adolescent educational level (between 2003 and 2013) [49]. A study of adolescents in Amsterdam between 2004 and 2013 found complex changes in socioeconomic inequalities by educational level, with increases in emotional symptoms in only the highest and lowest educational levels [50]. Since the effects of societal changes, such as the Great Recession, on socioeconomic inequalities in adolescent mental health may take time to emerge, it is important to look at longer-term changes and update previous research with the most recent data [51,52].

Using nationally representative samples of adolescents in the Netherlands from 2001 to 2017, three different SES indicators and six different indicators of adolescent mental health problems, this study examines three research questions regarding socioeconomic inequalities in adolescent mental health.
To what extent are three SES indicators—parental SES, adolescent subjective SES and adolescent educational level—independently associated with adolescent mental health problems?To what extent is the association between parental SES and adolescent mental health problems moderated by adolescent subjective SES and adolescent educational level?To what extent have the associations between three SES indicators, their interplay and adolescent mental health problems changed between 2001 and 2017?

## 2. Materials and Methods 

### 2.1. Participants

We used data from the Dutch Health Behaviour in School-aged Children (HBSC) study, a cross-sectional survey conducted every four years [53,54]. The present study made use of the data collected in the 2001, 2005, 2009, 2013 and 2017 surveys. The HBSC study used identical sampling and survey procedures across the survey years to collect data from nationally representative samples of 11- to 16-year-old adolescents attending the first four classes of secondary education (*N* = 28,310). A two-stage random cluster sampling procedure was used to obtain the samples. First, a random sample of schools in the Netherlands was drawn, and these were stratified based on urbanisation level. Second, each participating school provided a list of all classes, and 2–5 classes were selected randomly (depending on school size). All students were drawn as a single cluster within the selected classes, to ensure national representativeness survey weights were applied to the data. The response rate of schools ranged from 37% (2013/2017) to 48% (2009). The adolescent response rate was above 92% in all years. Self-report questionnaires (paper-and-pencil from 2001 to 2013 and computer-assisted in 2017) were administered in the classroom, taking ~40–50 min. The surveys took place in October or November of the corresponding year. Participants were informed of their anonymity and participants and parents gave passive consent (participants gave active consent in 2017). Ethical approval was gained from the Ethics Assessment Committee of the Faculty of Social Sciences at Utrecht University (FETC17-079 in 2017). 

Participants were included in the study if they had complete data on all SES measures, as well as gender and age, and reported at least one mental health problem measure (*n* = 27,020, over 95% of the full sample). The size of the included sample was similar across years: 2001 (*n* = 5277), 2005 (*n* = 5252), 2009 (*n* = 5321), 2013 (*n* = 5167) and 2017 (*n* = 6003). The mean age in every year was 13.8 (except in 2017, for which it was 13.7). Roughly half the sample was female (ranging from 49.7% in 2013 to 51.5% in 2017). 

### 2.2. Measures

#### 2.2.1. Socioeconomic Status Indicators

Parental SES was measured using the Family Affluence Scale (FAS) [55]. The scale consisted of 4 items that indicated family material assets: “Does your family have a car or a van?” (0 = no, 1 = yes, one, 2 = yes, two or more); “Do you have your own bedroom for yourself?” (0 = no, 1 = yes); “How many times did you travel away for holiday/vacation last year?” (0 = not at all, 1 = once, 2 = twice, 3 = more than twice; for the 2013 and 2017 HBSC study “away” was replaced with “abroad”); “How many computers does your family own?” (0 = none, 1 = one, 2 = two, 3 = more than two). Sum scores (range: 0–9) were computed for participants who completed all scale items; higher scores indicated more material assets. FAS, developed for use in the HBSC study, has been shown to be a reliable instrument that is easily answered by adolescents [55]. FAS is a composite indicator, constructed of separate independent items indicating family material assets, so internal consistency between items is not necessary [56]. We compared each adolescent’s absolute FAS sum score to all other scores in the respective survey year and calculated a ridit-based relative score, ranging from 0 to 1, with a mean of 0.5 [57]. By transforming the distribution to relative parental SES per year, we were able to minimise the effect of changes over time in the measure and meaning of FAS items [58,59].

Adolescent subjective SES was measured using the question, “How rich do you think your family is?” The item included a 5-point response scale (1 = very rich, 2 = quite rich, 3 = average, 4 = not so rich, 5 = not rich at all). The scale was reversed so that higher scores indicated higher levels of subjective SES. The measure has been found to be easy to answer for adolescents [54].

Adolescent educational level was measured using a question on the academic track that the adolescent followed. The secondary education system in the Netherlands has four tracks, and responses were dummy coded, with low used as the reference group in our analyses: 1 = low; 2 = lower intermediate; 3 = upper intermediate; 4 = high.

#### 2.2.2. Adolescent Mental Health Problems

Mental health is considered a multidimensional construct, with positive and negative indicators that are partly independent of one another [60]. As well as examining five indicators of mental health problems (emotional symptoms, conduct problems, hyperactivity, peer problems and psychosomatic complaints), we also include one indicator of positive mental health (life satisfaction).

Emotional symptoms, conduct problems, hyperactivity and peer problems were measured with the SDQ-R: a revised version of the problem subscales of the Strengths and Difficulties Questionnaire (SDQ), which has better psychometric properties than the original self-report SDQ [61]. From 2005 onwards, adolescents completed the Dutch translation of the self-report version of SDQ, which is suitable for 11- to 17-year-olds [62,63]. The SDQ comprises questions on behaviour and feelings over the past six months with 3-point response scales (0 = not true, 1 = somewhat true, 2 = certainly true). Examples of items are “Other children or young people pick on me or bully me” and “I get very angry and often lose my temper.” The SDQ-R consists of 15 items measuring four subscales—emotional symptoms (5 items), conduct problems (4 items), hyperactivity–inattention problems (3 items) and peer relationship problems (3 items) [61,62]. Two subscales—emotional symptoms (Cronbach’s α = 0.69–0.71) and hyperactivity–inattention problems (α = 0.69–0.74)—had acceptable internal consistency in all survey years, and although the other two subscales had lower α coefficients—conduct problems (α = 0.53–0.56) and peer relationship problems (α = 0.42–0.47)—these coefficients were consistent with existing research on the SDQ [61,64]. Previous research has indicated that the self-report SDQ is measurement invariant over time in the Netherlands [49]. Subscale mean scores were computed for participants who completed more than half of the subscale items. Higher subscale scores indicated more problems (ranging from 0 to 10).

Psychosomatic complaints were measured using the HBSC-symptom checklist. The checklist consists of eight items measuring: headache, abdominal pain, backache, dizziness, feeling low, irritability/bad temper, feeling nervous and sleeping difficulties. Each item included a 5-point response scale for reporting how often during the past 6 months the complaint was experienced (1 = about every day to 5 = rarely or never). To ensure that higher scores indicated more problems, we subtracted each score from 5 and computed a composite sum score for participants who completed at least six of the eight subscale items (ranging from 0 to 32) [65]. The checklist had a good internal consistency in all survey years (α = 0.76–0.82) and convergent validity with indicators for emotional symptoms [66].

Life satisfaction was measured using the Cantril Ladder [67]. The scale is an 11-point ladder with steps for reporting how participants feel about their life (0 = worst possible life to 10 = best possible life). The Cantril Ladder is easily understood and has shown high reliability among adolescents [68].

#### 2.2.3. Other Variables

Given gender and age differences in adolescent mental health problems, we controlled for these variables [3]. Gender was measured by asking whether the participant was a girl (coded 0) or boy (coded 1). Age was measured with a question about month and year of birth, which was used to calculate age at the date of data collection. Survey year was recoded to facilitate interpretation (2001 = 1, 2005 = 2, 2009 = 3, 2013 = 4, 2017 = 5) and included in models as a continuous variable.

### 2.3. Strategy of Analyses

To examine socioeconomic inequalities in adolescent mental health in the Netherlands, we tested several linear regressions separately for the six mental health outcomes: emotional symptoms, conduct problems, hyperactivity, peer problems, psychosomatic complaints and life satisfaction. Analyses were conducted using the complex samples module in SPSS (version 24, Armonk, NY, USA), so as to weight the data to ensure national representativeness and account for clustering within schools. All variables were standardised so that mean differences between groups (i.e., for gender and educational level) could be interpreted using Cohen’s *d* guidelines (small = 0.2, medium = 0.5, large = 0.8), and other effects could be interpreted as correlation coefficients (small = 0.1, medium = 0.3, large = 0.5) [69]. Given the large sample size and large number of tests, effects were considered significant when *p* < 0.001.

First, we included the control variables (M1), which were also included in all subsequent models. Second, we included the three SES indicators—(a) parental SES, (b) adolescent subjective SES and (c) adolescent educational level—separately (M2a-c). Third, we included all three SES indicators simultaneously (M3). Fourthly, we included all three SES indicators and a two-way interaction between (a) parental SES and adolescent subjective SES, or (b) parental SES and adolescent educational level (M4a-b). Fifth, we included survey year (5a), then all three SES indicators and three two-way interactions between survey year and each SES indicator (M5b). Sixth, we included all three SES indicators; two-way interactions between survey year and each SES indicator and three-way interactions between (a) survey year, parental SES and adolescent subjective SES, or (b) survey year, parental SES and educational level (M6a-b). According to a power analysis with the software G*Power, based on an *α* level of 0.001 and a desired power (1–*β*) of 0.85, a three-way interaction effect size of *R*² = 0.001 can be detected with a sample of 20,910 adolescents [70].

## 3. Results

### 3.1. Descriptive Statistics

We used nationally representative self-reported data from 11- to 16-year-old adolescents participating in the Dutch Health Behaviour in School-Aged Children (HBSC) studies in 2001, 2005, 2009, 2013 and 2017 (*n* = 27,020; see Methods). Table 1 shows changes within the sample in the mean and standard deviation of SES and adolescent mental health problems between 2001 and 2017. The parental SES score was designed to have the same mean for each survey year and mean levels of adolescent subjective SES did not change substantially over time. During the period, there was a 5.4 percentage point decrease in Dutch adolescents in the low educational level and an 8.5 percentage point increase in participation in the high educational level. Mean levels of the mental health problems were quite low, except for hyperactivity, which was moderate, whereas life satisfaction was quite high.

Table 2 shows that all three SES indicators had small positive associations with each other (*r*s range from 0.11 to 0.38), indicating that they measured different dimensions of SES that may have an underlying component. Mental health problems were positively associated with each other (*r*s range from 0.11 to 0.63) and negatively associated with life satisfaction (*r*s range from 0.19 to 0.46). Associations were mostly small in size (|*r*|< 0.30), suggesting that the indicators measured different dimensions of mental health problems that may have an underlying component.

Table 3 shows the results of the linear regression models examining socioeconomic inequalities in adolescent mental health. Model 1, which included only control variables, showed age and gender differences in adolescent mental health problems. Boys reported higher levels of conduct problems and higher life satisfaction, though the effects were small. Girls reported higher levels of emotional symptoms and psychosomatic complaints; the effects were medium-sized. Gender differences in peer problems and hyperactivity were absent or of negligible size. Age differences in mental health problems were negligible, but younger adolescents reported somewhat higher levels of life satisfaction.

### 3.2. Associations between Three SES Indicators and Adolescent Mental Health Problems

Models 2a–c examined all SES indicators separately. The results showed that, in general, all three SES indicators—parental SES, adolescent subjective SES and adolescent educational level—were negatively associated with mental health problems and positively associated with life satisfaction. Apart from the medium-sized negative association between a high educational level and conduct problems, other associations were negligible to small in size. Model 3, which included all three SES indicators simultaneously, showed that adolescent subjective SES and adolescent educational level were independently associated with adolescent mental health problems: adolescent subjective SES was negatively associated with emotional symptoms and positively associated with life satisfaction, and higher educational levels were negatively associated with conduct problems, hyperactivity and peer problems. Associations between parental SES and adolescent mental health problems attenuated (except between parental SES and hyperactivity, which increased) and all associations were negligible in size.

### 3.3. Moderation Effects of Adolescent Subjective SES and Adolescent Educational Level

Model 4a shows that adolescent subjective SES moderated the association between parental SES and adolescent mental health problems and life satisfaction for all outcomes except peer problems. To facilitate interpretation, we depicted the associations between parental SES and adolescent mental health problems and life satisfaction for two values of adolescent subjective SES: low (one standard deviation below the mean) and high (one standard deviation above the mean; Figure 1). Among adolescents with low adolescent subjective SES, parental SES was negatively associated with emotional symptoms, conduct problems and psychosomatic complaints, and positively associated with life satisfaction. However, among adolescents with high adolescent subjective SES, the association between parental SES and these mental health outcomes was not significant (or, in the case of psychosomatic complaints and conduct problems, positive). Thus, for emotional symptoms, conduct problems and psychosomatic complaints, high adolescent subjective SES buffered the association between lower parental SES and higher levels of adolescent mental health problems. High adolescent subjective SES also buffered the association between lower parental SES and lower life satisfaction.

For hyperactivity, the positive association between parental SES and hyperactivity was stronger when adolescent subjective SES was high than when adolescent subjective SES was low. Thus, high adolescent subjective SES amplified the association between higher parental SES and higher levels of hyperactivity.

Model 4b shows that there were no significant interaction effects between parental SES and adolescent educational level on mental health problems and life satisfaction. Thus, adolescent educational level did not moderate the association between parental SES and adolescent mental health problems and life satisfaction.

### 3.4. Changes in Socioeconomic Inequalities between 2001 and 2017

Model 5a shows that there was a negligible increase in emotional symptoms and a small increase hyperactivity (between 2005 and 2017) and a small increase in psychosomatic complaints and a negligible decrease in life satisfaction (between 2001 and 2017). Model 5b shows no significant changes in the associations between the SES indicators and adolescent mental health problems and life satisfaction over time, and Models 6a and 6b show no significant changes in the moderation effects of adolescent subjective SES and adolescent educational level on the association between parental SES and adolescent mental health problems and life satisfaction. As a sensitivity analysis, we also ran models with survey year as a categorical variable, but the results were not substantially different.

## 4. Discussion

By including three different SES indicators and six different mental health outcomes, the results of the present study shed light on socioeconomic inequalities in adolescent mental health in the Netherlands. The findings showed that low adolescent subjective SES and low adolescent educational level had small independent associations with adolescent mental health problems. Independent associations between parental SES and adolescent mental health problems were negligible. Furthermore, for four out of six mental health outcomes, high subjective SES buffered the negative association between parental SES and adolescent mental health problems. In addition, this study found no changes in the associations between the SES indicators and adolescent mental health problems between 2001 and 2017, indicating persistent socioeconomic inequalities in adolescent mental health despite societal changes.

### 4.1. Adolescent Subjective SES and Educational Level Associated with Mental Health Problems

We replicated previous findings in relatively wealthy and egalitarian countries, showing that the small association between parental SES and adolescent mental health problems attenuates when including other SES indicators [32,71]. In such countries, like the Netherlands, where income is redistributed and there are universal public services, the association between parental SES and mental health problems appears to be very small [8,13].

In accordance with previous studies, we found that the association between adolescent subjective SES and adolescent mental health problems is stronger than the association between parental SES and adolescent mental health problems [7,72,73]. Perceptions of status appear to be especially important during the adolescent period [74]. In particular, we found that lower adolescent subjective SES was associated with more emotional symptoms and lower life satisfaction. Our findings that associations are strongest with internalising problems suggest that social comparison effects, perhaps eliciting shame and pessimism, may be important mechanisms in explaining the association [19,20,21]. The association between subjective SES and mental health problems may also be due to reverse causation, because mental health problems may cause adolescents to perceive their environment less positively and their family as less wealthy [75,76]. The association may also be confounded by a third variable; personality traits, such as self-esteem, could affect how individuals rate themselves in general and influence their reports of subjective SES and mental health problems [19]. However, several experimental studies (conducted with adult populations) suggest that low subjective SES does cause mental health problems [19,77]. To further understand the association between subjective SES and adolescent internalising mental health problems and life satisfaction, it would be useful to further study which factors cause adolescent’s subjective SES evaluations [7].

Educational level, an indicator of emerging adolescent SES, was also independently associated with adolescent mental health problems, with low education associated with externalising problems (in particular, conduct problems as well as hyperactivity) and peer problems. These results support previous findings from the Netherlands and other egalitarian countries [13,32]. As our results are strongest for conduct problems and negligible for internalising problems and life satisfaction, it is more likely that adolescents with lower educational levels have more mental health problems because they experience a negative classroom climate and poor peer behaviour, rather than feelings of academic inadequacy and inferiority [24,25,26]. As with subjective SES, causation could be in the opposite direction; adolescents with mental health problems may struggle to cope in school and be tracked into a low educational level [78,79]. Additionally, the association between a low adolescent educational level and mental health problems may be due to genetic factors or heritable traits [80]. Given the associations we found, research on socioeconomic inequalities in adolescent mental health should take into account the role of adolescent educational level, and consider factors other than educational level which may also indicate emerging adolescent SES, such as educational or occupational expectations [12,22,81]. Further research could also explore whether the associations found depend on age; parental SES may be a stronger predictor of childhood mental health, but have a decreasing impact as adolescents acquire their own SES [3].

### 4.2. Adolescent Subjective SES Buffers the Association between Low Parental SES and Adolescent Mental Health Problems

Although parental SES did not have an independent association with adolescent mental health problems when adolescent subjective SES and educational level were included, we found that high subjective SES buffered the negative association between parental SES and mental health problems for four of six outcomes. The results suggest that in wealthy and egalitarian countries such as the Netherlands, high subjective SES may be a protective factor against mental health problems for adolescents with lower parental SES [21]. High subjective SES, perceiving one’s family situation favourably, appears to be especially important to adolescents who experience adverse circumstances of lower parental SES [21]. For the fifth outcome, hyperactivity, high adolescent subjective SES amplified a negligible, but unexpected, positive association between parental SES and hyperactivity. In the Netherlands, where adolescents report relatively high levels of hyperactivity compared to their European agemates [61], hyperactivity may be more normative than problematic [82], contrasting with research findings elsewhere that higher levels of hyperactivity are typically found in lower SES families [83,84]. Our findings are correlational, thus further research on the interaction between parental SES and adolescent subjective SES could examine possible underlying mechanisms, such as psychosocial resources (e.g., optimism) and perceived social support [21,85,86]. Understanding of these mechanisms could be important for developing effective ways to reduce socioeconomic inequalities in adolescent mental health.

We did not find a moderation effect of educational level on the association between parental SES and adolescent mental health problems. Perhaps, in a relatively wealthy and egalitarian country with a tracked education system, the associations of parental SES and educational level with adolescent mental health problems are cumulative due to the intergenerational transmission of educational level [87]. Furthermore, theories that adolescents in higher educational levels with lower parental SES may experience mental health benefits—gaining cognitive skills that can be used to reduce the effects of stress [34]—or mental health disadvantages—experiencing feelings of social isolation or alienation [36]—may not be so relevant to the Netherlands, where lower parental SES may not be so different from higher parental SES.

### 4.3. No Changes in Socioeconomic Inequalities in Adolescent Mental Health

Our results showed socioeconomic inequalities in adolescent mental health did not change between 2001 and 2017. This finding was remarkably consistent—holding for all three SES indicators and all six mental health outcomes, despite societal changes during the period. The results extend the findings of two previous studies, which covered shorter timespans, but also showed stable socioeconomic inequalities in adolescent mental health in the Netherlands using one SES indicator [48,49]. Socioeconomic inequalities in adolescent mental health appear to be stable in several other wealthy countries as well, including Sweden and Canada [11,66,71]. Perhaps, due to living in a country with a strong welfare state [88], changes in the economy and society—including the Great Recession, educational changes and the growth of digital media—have not substantially affected the proximal environment of adolescents, which is most likely to contribute to their mental health [11,89]. However, our results also showed that socioeconomic inequalities in adolescent mental health persist, thus even with the Netherlands’ strong welfare state, policies to reduce these inequalities do not appear to be especially effective [33].

### 4.4. Strengths and Limitations

The large, nationally representative HBSC dataset, with consistency in the survey questions and sampling procedures across a 16-year time span enabled us to include multiple SES indicators and mental health measures, consider time trends and use data from tens of thousands of adolescents, which provides confidence about the robustness of our results. However, this study also has several limitations. First, differences in the characteristics of the five samples (other than gender and age, which we control for), such as potential changes over time in family structure, may be responsible for the consistent association between the SES indicators and adolescent mental health problems over time. Nevertheless, by using national representative data at all time points, we are able to assess how SES associations with mental health problems have changed in the general adolescent population in the Netherlands. Secondly, our measure of parental SES was designed for cross-national research and is skewed towards high parental SES in the Dutch sample, though by using a relative measure we were able to lessen this limitation [2]. Further research could explore whether other measures of parental SES (such as parental income or occupation) have similar associations with adolescent mental health problems and if they interact with adolescent subjective SES. Thirdly, the cross-sectional nature of the data does not enable us to draw causal conclusions. As discussed, mental health may partly influence both subjective SES and educational level, or share (genetic) factors which confound the association. Fourthly, we did not have data for SDQ-R measured mental health problems in 2001, though this was mitigated by our lack of findings of changes in psychosomatic complaints and life satisfaction for the full time-period studied. Fifthly, our results for early adolescents (aged 11–16) in the Netherlands may not generalise to older adolescents or adolescents living in other countries, though many of our findings support existing evidence with different samples.

## 5. Conclusions

By looking at parental SES, adolescent subjective SES, and adolescent educational level and their interplay, this study adds important new insights to the literature on socioeconomic inequalities in adolescent mental health. We replicated previous findings showing that lower adolescent subjective SES and educational level were independently associated with adolescent mental health problems, but parental SES was not. Findings differed by mental health outcome: adolescent subjective SES was associated with internalising problems and life satisfaction, while adolescent educational level was associated with externalising and peer problems. Extending the existing studies, we found that, when interactions between SES indicators were considered, high subjective SES buffered adolescents from the mental health problems associated with having low parental SES. Socioeconomic inequalities in adolescent mental health remained stable for 16 years. Taken together, these findings suggest that researchers, policy-makers and practitioners looking to improve adolescent mental health and reduce socioeconomic inequalities need to consider all three SES indicators—parental SES, adolescent subjective SES and adolescent educational level—to effectively target interventions towards adolescents most likely to experience mental health problems.

## Figures and Tables

**Figure 1 ijerph-16-03605-f001:**
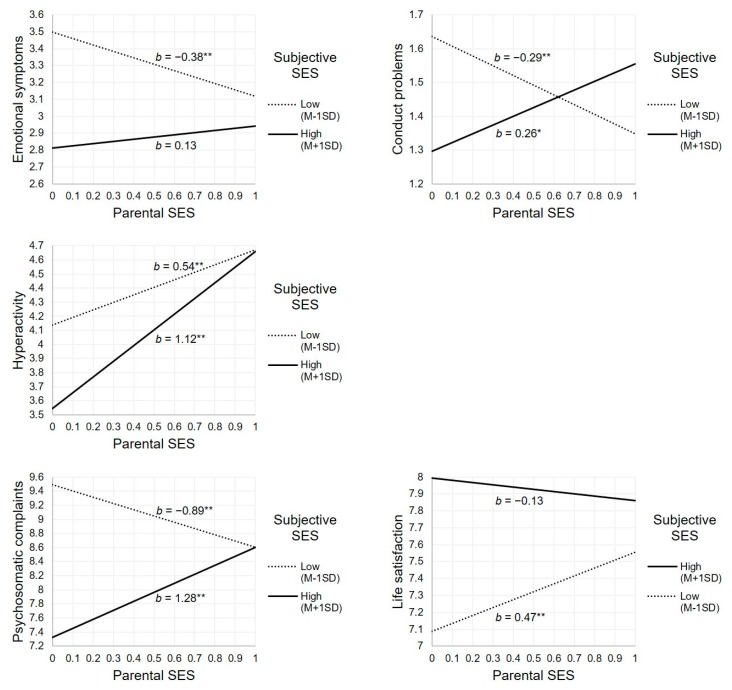
Conditional effect of parental SES on adolescent mental health for two values of adolescent subjective SES (Age = 13.9; Gender = female; Educational level = low). * *p* < 0.001, ** *p* < 0.0001.

**Table 1 ijerph-16-03605-t001:** Descriptive statistics for socioeconomic status (SES) and mental health problems by year.

	Range	2001	2005	2009	2013	2017	Total	Total *n*
Parental SES (*M*/*SD*)	0–1	0.5 (0.3)	0.5 (0.3)	0.5 (0.3)	0.5 (0.3)	0.5 (0.3)	0.5 (0.3)	27,020
Subjective SES (*M*/*SD*)	1–5	3.3 (0.7)	3.1 (0.7)	3.2 (0.7)	3.1 (0.8)	3.2 (0.7)	3.2 (0.7)	27,020
Low Educational Level (%)		23.5	26.0	17.8	23.3	18.0	21.6	27,020
Lower Int. Educational Level (%)		35.4	31.5	35.5	27.1	32.3	32.4	27,020
Upper Int. Educational Level (%)		23.1	24.4	22.9	26.8	23.1	24.0	27,020
High Educational Level (%)		18.0	18.1	23.8	22.8	26.5	22.0	27,020
Emotional Symptoms (*M*/*SD*)	0–10		2.2 (2.1)	2.2 (2.1)	2.6 (2.3)	2.5 (2.3)	2.4 (2.2)	21,503
Conduct Problems (*M*/*SD*)	0–10		1.3 (1.8)	1.1 (1.7)	1.2 (1.7)	1.2 (1.7)	1.2 (1.7)	21,480
Hyperactivity (*M*/*SD*)	0–10		3.6 (2.8)	3.9 (2.9)	4.1 (3.0)	4.1 (3.0)	4.0 (2.9)	21,518
Peer Problems (*M*/*SD*)	0–10		1.7 (1.9)	1.7 (2.0)	1.8 (2.0)	1.9 (2.0)	1.8 (2.0)	21,484
Psychosomatic Complaints (*M*/*SD*)	0–32	6.6 (5.6)	5.6 (5.8)	6.5 (5.6)	7.9 (6.4)	7.9 (6.5)	6.9 (6.1)	26,813
Life Satisfaction (*M*/*SD*)	0–10	7.9 (1.6)	7.7 (1.6)	7.9 (1.4)	7.6 (1.6)	7.6 (1.6)	7.7 (1.6)	26,729

Note. Int. = intermediate.

**Table 2 ijerph-16-03605-t002:** Correlations between variables.

	Variables	2	3	4	5	6 *^b^*	7	8	9	10	11	12
Control	1. Age	0.02	−0.03 **	−0.06 **	−0.05 **	−0.10 **	0.06 **	0.02 *	0.05 **	0.07 **	0.06 **	−0.15 **
	2. Gender *^a^*		−0.01	0.10 **	0.11 **	−0.01	−0.31 **	0.12 **	0.01	0.03 **	−0.22 **	0.14 **
Time	3. Survey Year			0.00	−0.03 **	0.08 **	0.06 **	−0.01	0.07 **	0.03 **	0.12 **	−0.08 **
SES	4. Parental SES				0.38 **	0.21 **	−0.09 **	−0.04 **	0.03 **	−.11 **	−0.05 **	0.12 **
	5. Adol. Subjective SES					0.11 **	−0.14 **	−0.04 **	−0.04 **	−0.08 **	−0.12 **	0.23 **
	6. Adol. Educ. Level *^b^*						−0.02 **	−0.20 **	−0.11 **	−0.15 **	−0.01	0.01
Adol. Mental Health Problems	7. Emotional Symptoms							0.28 **	0.29 **	0.35 **	0.63 **	−0.46 **
	8. Conduct Problems								0.33 **	0.29 **	0.31 **	−0.23 **
	9. Hyperactivity									0.11 **	0.31 **	−0.19 **
	10. Peer Problems										0.26 **	−0.23 **
	11. Psychosomatic Complaints											−0.45 **
	12. Life Satisfaction											

Note. *^a^* Female is the reference group. *^b^* Correlation coefficients with educational level are Spearman’s (all other correlations are Pearson’s). Adol. = adolescent; educ. = educational. ns vary due to missing data, ranging from 21,239 to 27,020. * *p* < 0.001, ** *p* < 0.0001.

**Table 3 ijerph-16-03605-t003:** Regression models 1–6b showing the effects of SES indicators on six adolescent mental health problems.

		Emotional Symptoms		Conduct Problems		Hyperactivity		Peer Problems		Psychosomatic Complaints		Life Satisfaction	
		*B* (SE)	*β*	*B* (SE)	*β*	*B* (SE)	*β*	*B* (SE)	*β*	*B* (SE)	*β*	*B* (SE)	*β*
M1	Intercept	1.67 ** (0.18)	0.00	0.57 ** (0.17)	0.01	2.50 ** (0.29)	0.00	0.26 (0.18)	0.00	4.29 ** (0.49)	0.00	10.14 ** (0.12)	0.00
	Gender (girls = ref)	−1.39 ** (0.03)	−0.31	0.43 ** (0.03)	0.12	0.08 (0.04)	0.01	0.14 ** (0.03)	0.04	−2.71 ** (0.08)	−0.22	0.44 ** (0.02)	0.14
	Age	0.10 ** (0.01)	0.06	0.03 (0.01)	0.02	0.10 ** (0.02)	0.04	0.11 ** (0.01)	0.07	0.29 ** (0.04)	0.06	−0.19 ** (0.01)	−0.15
M2a	Parental SES	−0.43 ** (0.06)	−0.06	−0.36 ** (0.05)	−0.06	0.33 * (0.10)	0.03	−0.80 ** (0.06)	−0.11	−0.66 ** (0.16)	−0.03	0.55 ** (0.04)	0.10
M2b	Adol. Subjective SES	−0.33 ** (0.02)	−0.11	−0.12 ** (0.02)	−0.05	−0.16 ** (0.03)	−0.04	−0.22 ** (0.02)	−0.08	−0.77 ** (0.06)	−0.09	0.44 ** (0.02)	0.21
M2c	Educ. = High	−0.19 (0.06)	−0.08	−0.98 ** (0.05)	−0.57	−0.88 ** (0.09)	−0.30	−0.78 ** (0.05)	−0.40	−0.71 * (0.20)	−0.12	0.07 (0.04)	0.05
	Educ. = Upper Int.	−0.08 (0.06)	−0.04	−0.69 ** (0.05)	−0.40	−0.29 * (0.10)	−0.10	−0.56 ** (0.05)	−0.29	−0.54 (0.19)	−0.09	−0.02 (0.04)	−0.02
	Educ. = Lower Int.	−0.12 (0.06)	−0.05	−0.37 ** (0.05)	−0.21	−0.10 (0.09)	−0.03	−0.27 ** (0.06)	−0.13	−0.37 (0.18)	−0.06	−0.04 (0.04)	−0.03
M3	Parental SES	−0.12 (0.06)	−0.02	−0.01 (0.05)	0.00	0.83 ** (0.10)	0.08	−0.49 ** (0.06)	−0.07	0.25 (0.16)	0.01	0.15 * (0.04)	0.03
	Adol. Subjective SES	−0.31 ** (0.03)	−0.10	−0.06 (0.02)	−0.02	−0.22 ** (0.03)	−0.05	−0.11 ** (0.02)	−0.04	−0.78 ** (0.07)	−0.09	0.42 ** (0.02)	0.20
	Educ. = High	−0.09 (0.07)	−0.04	−0.97 ** (0.05)	−0.56	−0.98 ** (0.08)	−0.34	−0.66 ** (0.05)	−0.034	−0.57 (0.20)	−0.09	−0.05 (0.04)	−0.03
	Educ. = Upper Int.	−0.03 (0.06)	−0.01	−0.68 ** (0.05)	−0.39	−0.37 ** (0.09)	−0.13	−0.48 ** (0.05)	−0.25	−0.48 (0.19)	−0.08	−0.09 (0.04)	−0.06
	Educ. = Lower Int.	−0.10 (0.06)	−0.04	−0.37 ** (0.05)	−0.21	−0.14 (0.09)	−0.05	−0.23 ** (0.06)	−0.12	−0.35 (0.18)	−0.06	−0.07 (0.04)	−0.04
M4a	Parental SES × Adol. Subjective SES	0.35 ** (0.08)	0.03	0.37 ** (0.07)	0.05	0.39 * (0.11)	0.03	0.22 (0.08)	0.02	1.48 ** (0.20)	0.05	−0.41 ** (0.05)	−0.05
M4b	Parental SES × Educ. = High	−0.40 (0.18)	−0.05	−0.25 (0.13)	−0.04	−0.32 (0.23)	−0.03	−0.42 (0.17)	−0.06	0.15 (0.49)	0.01	−0.05 (0.12)	−0.01
	Parental SES × Educ. = Upper Int.	−0.33 (0.18)	−0.04	−0.12 (0.13)	−0.02	−0.07 (0.24)	−0.01	−0.29 (0.17)	−0.04	0.37 (0.49)	0.02	−0.16 (0.12)	−0.03
	Parental SES × Educ. = LOWER Int.	−0.35 (0.18)	−0.04	−0.17 (0.14)	−0.03	0.16 (0.23)	0.02	−0.39 (0.16)	−0.06	0.13 (0.48)	0.01	−0.12 (0.12)	−0.02
M5a	Survey Year	0.12 ** (0.02)	0.08	−0.02 (0.02)	−0.02	0.20 ** (0.03)	0.10	0.06 (0.02)	0.04	0.50 ** (0.04)	0.12	−0.09 ** (0.01)	−0.08
M5b	Year × Parental SES	−0.14 (0.05)	−0.03	−0.05 (0.04)	−0.01	0.05 (0.09)	0.01	−0.13 (0.05)	−0.03	0.10 (0.12)	0.01	0.02 (0.03)	0.01
	Year × Adol. Subjective SES	−0.02 (0.02)	−0.01	−0.05 (0.02)	−0.03	−0.07 (0.03)	−0.02	−0.03 (0.02)	−0.02	−0.11 (0.04)	−0.02	0.00 (0.01)	0.00
	Year × Educ. = High	0.05 (0.05)	0.03	0.10 (0.04)	0.08	−0.06 (0.07)	−0.03	0.04 (0.05)	0.03	0.06 (0.12)	0.01	0.00 (0.03)	0.00
	Year × Educ. = Upper Int.	0.06 (0.05)	0.04	0.11 (0.04)	0.09	0.01 (0.07)	0.00	0.05 (0.04)	0.04	0.26 (0.12)	0.06	−0.04 (0.03)	−0.03
	Year × Educ. = Lower Int.	0.04 (0.05)	0.03	0.05 (0.04)	0.05	0.00 (0.08)	0.00	0.03 (0.05)	0.02	0.08 (0.11)	0.02	−0.03 (0.03)	−0.03
M6a	Year × Parental SES × Adol. Subjective SES	−0.07 (0.07)	−0.01	−0.01 (0.07)	0.00	−0.02 (0.09)	0.00	−0.01 (0.07)	0.00	−0.08 (0.13)	0.00	−0.01 (0.04)	0.00
M6b	Year × Parental SES × Educ. = High	0.20 (0.16)	0.04	0.06 (0.11)	0.01	0.34 (0.19)	0.05	−0.04 (0.14)	−0.01	0.62 (0.34)	0.04	−0.04 (0.08)	−0.01
	Year × Parental SES Educ. = Upper Int.	0.26 (0.16)	0.05	0.20 (0.12)	0.05	0.19 (0.21)	0.03	0.13 (0.14)	0.03	0.35 (0.34)	0.02	−0.03 (0.08)	−0.01
	Year × Parental SES Educ. = Lower Int.	0.06 (0.16)	0.01	−0.10 (0.12)	−0.02	0.03 (0.21)	0.00	−0.17 (0.12)	−0.03	−0.22 (0.32)	−0.01	0.04 (0.08)	0.01

Note. Adol. = adolescent; Educ. = adolescent educational level; Int. = intermediate. Educ. = low (and interactions) are reference category. * *p* < 0.001, ** *p* < 0.0001.

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
