# Peer review of "Adolescent Socioeconomic Status and Mental Health Inequalities in the Netherlands, 2001–2017"

_ijerph, 2019, doi:10.3390/ijerph16193605_

Round 1
Reviewer 1 Report
The authors attempted to investigate the interplay between three SES indicators - parental SES, adolescent subjective SES, and adolescent educational level - and adolescent mental health problems and explore changes in these associations during the last two decades. However, the authors have not had total success in achieving their objectives. In fact, I have some comments.
1.- The authors use cross-sectional data collected in the 2001, 2005, 2009, 2013 and 2017 surveys. This type of design is accompanied by the ‘Evaluation problem’. This problem occurs when an event is evaluated using data in the form of a cross section. The problem is that the subjects interviewed before and after the event are not the same ones, thus making the groups (of each cross section) not comparable.
If the evaluation problem is not controlled, the results could be wrong. The control can be done without changing the design, which is equivalent to controlling confounding and three strategies could be followed to do so: Matching, stratification, and adjustment in a multivariate model.
The authors have opted for a multivariate model. The problem is that they have included very few confusers (age and gender of the adolescent and survey year). The problem is that there may be other confusers that have not been controlled. In particular some observed and others not observed.
The authors should, not so much re-analyze the models, as to discuss, in detail, the evaluation problem, the possible residual confusion, etc. and its effects on the results.
2.- The authors include Survey year as a continuous variable. Perhaps they should include it as a categorical variable, in order to control this confounder each year separately.
Author Response
1. We thank the reviewer for alerting us to this issue and have added a section to the limitations to address it.
Discussion – addition (page 12)
Firstly, differences in the characteristics of the five samples (other than gender and age, which we control for), such as potential changes over time in family structure, may be responsible for the consistent association between the SES indicators and adolescent mental health problems over time, Nevertheless, by using national representative data at all time points, we are able to assess how SES associations with mental health problems have changed in the general adolescent population in the Netherlands.
2. By using survey year as a continuous variable, we assessed linear changes in the association between SES and adolescent mental health during the period from 2001-2017. Using a categorical year variable, we can only compare pairs of years. We have added a line in the results to note that running the analyses with survey year as a categorical variable does not substantially change our results.
Results – addition (page 10)
As a sensitivity analysis, we also ran models with survey year as a categorical variable, but the results were not substantially different.
Reviewer 2 Report
In general, this is an interesting and well executed research paper. However, I think the authors can improve its quality if the following issues are taken into account:
- I think it is right to consider “life satisfaction” as an indicator of mental health, but not as a problem. In the title and keywords, the authors correctly say only "mental health", but in the rest of the document they group all the dependent variables as "mental health problems" (except on line 260: "they were negatively associated with mental health problems and positively with life satisfaction"). Therefore, I suggest to have this consideration in the rest of the document, in the way that the authors consider best.
- It would be appropriate to specify in the method the sample size of each edition (2001-2005-2009-2013-2017).
- Authors must provide some reliability indicator for the FAS and SDQ scales. In the latter case, at least provide reliability for the dimensions independently, since this is the use that is given to the variable SDQ.
- There is a mistake on line 203: not say “2019 = 5” but “2017 = 5”.
- Line 211: d must be italicized (Cohen’s d).
- To properly interpret the means of table 1 it is necessary to know the minimum and maximum values of each variable. This information is in the method section, but it would be appropriate to ensure that it is clearly identified in all the variables (for example, in the variable psychosomatic).
- Lines 267 - 268 (“Associations between parental SES and adolescent mental health problems attenuated and became negligible in size”): it would be good to specify more these results, that is, to indicate in which mental health variables the results are attenuated and in which they become negligible. And, to apply this also in the discussion.
Round 2
Reviewer 1 Report
The authors have answered all my comments and have incorporated most of them into the new version of the manuscript. I have no further comments.